# User Participatory Design of a Wearable Focal Vibration Device for Home-Based Stroke Rehabilitation

**DOI:** 10.3390/s22093308

**Published:** 2022-04-26

**Authors:** Hongwu Wang, Mustafa Ghazi, Raghuveer Chandrashekhar, Josiah Rippetoe, Grace A. Duginski, Louis V. Lepak, Lisa R. Milhan, Shirley A. James

**Affiliations:** 1Department of Occupational Therapy, University of Florida, Gainesville, FL 32603, USA; rchandrashekhar@phhp.ufl.edu; 2Department of Rehabilitation Sciences, University of Oklahoma Health Sciences Center, Oklahoma City, OK 73117, USA; mustafa-ghazi@ouhsc.edu (M.G.); josiah-rippetoe@ouhsc.edu (J.R.); louis-lepak@ouhsc.edu (L.V.L.); lisa-milhan@ouhsc.edu (L.R.M.); shirley-james@ouhsc.edu (S.A.J.); 3Stephenson School of Biomedical Engineering, University of Oklahoma, Norman, OK 73019, USA; duginskiga@ou.edu

**Keywords:** user participatory design, wearable device, focal vibration, stroke rehabilitation, home-based rehabilitation

## Abstract

Stroke often leads to the significant impairment of upper limb function and is associated with a decreased quality of life. Despite study results from several interventions for muscle activation and motor coordination, wide-scale adoption remains largely elusive due to under-doses and low user compliance and participation. Recent studies have shown that focal vibration has a greater potential to increase and coordinate muscle recruitment and build muscle strength and endurance. This form of treatment could widely benefit stroke survivors and therapists. Thus, this study aimed to design and develop a novel wearable focal vibration device for upper limb rehabilitation in stroke survivors. A user participatory design approach was used for the design and development. Five stroke survivors, three physical therapists, and two occupational therapists were recruited and participated. This pilot study may help to develop a novel sustainable wearable system providing vibration-based muscle activation for upper limb function rehabilitation. It may allow users to apply the prescribed vibratory stimuli in-home and/or in community settings. It may also allow therapists to monitor treatment usage and user performance and adjust the treatment doses based on progression.

## 1. Introduction

Stroke is the second leading cause of death and the third leading cause of disability-adjusted life-years worldwide. Each year, approximately 795,000 people in the United States have strokes. Stroke is the leading cause of serious, long-term disability in the United States [1], and up to 85% of stroke survivors experience some degree of paresis of the upper limb [2]. Because most strokes affect the middle cerebral artery, less recovery is common in the upper limbs than in the lower limbs. Moreover, about 50% of stroke survivors experience chronic upper limb and hand function impairment [3]. Upper limb impairments can lead to functional limitations in reach, grasp, and manipulation, critical for completing basic activities of daily living (ADLs) such as self-care, eating/drinking, and meal preparation [4]. Limited upper limb function and ADL performance are key factors associated with the perception of a reduced quality of life [5,6] and poorer perception of well-being [7]. Therefore, improving upper limb function is a core element of rehabilitation after stroke [1].

While numerous therapies have been developed over the last ten years to treat acute ischemic stroke, the stark reality remains that 95% of these stroke survivors continue intervention in the chronic stage and go on to live with a significant disability for many years [8,9]. Improvements in upper limb function occur through various combinations of spontaneous recovery, remediation, and compensation [1,3]. Stroke survivors have difficulty moving out of the upper extremity flexion synergies that often dominate attempts to function after stroke [10]. Changes in the ability to activate muscles independently are the key underlying factors for these synergies. Despite promising study results from several new interventions for muscle activation and motor coordination, wide-scale adoption remains largely elusive due to the lack of sustainability of those interventions [11]. In the context of FV in this paper, sustainability of intervention is defined as being readily available and regularly available without interruption. In terms of users, the limiting factors are the limited number and frequency of clinical visits a user can make and the non-availability of the FV devices for personal use at home. In terms of the therapists, the limiting factors are the number of available therapists, their limited available time, and limited equipment availability to provide regular FV therapy. Together, these factors result in an under-dosage because the users cannot get FV frequently enough, regularly enough, or for a long enough period to have any meaningful effect. The main reason for the unsustainability is the under-dosage of the interventions [12]. The issue is exacerbated by low user compliance and participation because stroke recovery and rehabilitation often require multiple treatment sessions, each requiring a visit to the study site.

Vibration has, over time, gained an important role in physical and rehabilitative medicine. Mechanical perturbations of small amplitudes localized on specific muscles or tendons, known as focal vibration (FV), can easily be kept within safe limits for muscle activation. Previous studies have found that FV reduces muscle spasticity [13,14,15,16], facilitates muscle contraction [17,18], and stimulates the proprioceptive system to obtain efficient motor control during functional activities [19]. Studies show that FV can enhance sensorimotor organization through proprioceptive training in individuals with movement disorders [20,21]. Individuals with multiple sclerosis, dystonia, stroke, and incomplete spinal cord injuries have demonstrated improvements in gait cadence, step length, and walking speed after the consistent use of focal vibration [22]. Specifically, studies reported that FV increased the stability of the proximal arm in the execution of motor tasks [23] and improved the muscle activity registered with surface EMG [24] in the upper extremities of individuals who have had strokes. Several studies reported decreases in spasticity and functional gains from focal vibration [25,26,27,28,29,30,31,32]. FV applied to the forearm and arm flexor muscles significantly improved grip strength and scores on the box–and–block test [33]. Applying 300 Hz of FV to the triceps brachii and extensor carpi radialis longus and brevis led to improved muscle strength, decreased muscle tone, and decreased pain [34].

At present, clinical application and in-home rehabilitation using FV is rare due to the lack of uniform protocols and sustainability, even though the intervention may be easier to use, better tolerated, and more effective on muscle activation than traditional intervention. In a recent review study [35], vibration frequencies have ranged from 60 Hz to 300 Hz, and amplitudes have ranged from 0.01 mm to 10 mm. The duration of the treatment has varied from 5 min to 60 min. In addition, previous studies that investigated the long-term effect of FV still required frequent study visits due to the limitation of the FV devices used. FV delivered in a wearable format could be a sustainable home-based rehabilitation intervention. It allows users to get therapy without needing help from another person and without the need to hold up the device in often awkward and tiring poses. MyoVolt is a newly developed wearable FV device that allows some preset vibration settings but lacks compliance-monitoring capabilities [36]. There is also no alternative interface other than the on-device switches, so awkward reaching is required to operate the device in many cases. In our previous studies using MyoVolt for low extremity rehabilitation in individuals with peripheral neuropathy [37,38] and multiple sclerosis [39], it was promising that a wearable FV may improve mobility and function. Optimal dosage parameters still need investigation. Device usage requires a quantified approach that does not involve a participant remembering the dosage parameters, time durations, and schedules. Documentation of devise usage is complicated for those participants with cognitive impairments. Device documentation is essential so that recovery and compliance can be tracked. Another common complaint from the participants was the lack of adjustability and individualized vibration prescription. Finally, based on our engineer testing of different wearable focal vibration devices, most devices did not deliver consistent vibration, which is likely because of the design of the devices. The motor/skin interface and the tightness of the attachment mechanism can and do change the vibration delivered. Thus, the vibration levels delivered are not consistent. Therefore, in this study, we aimed to design, develop, and bench test a wearable device, FoVi, that applies accurate vibration to specific muscles and allows therapists to use an app to adjust the treatment and receive feedback in real-time along with usage.

## 2. Materials and Methods

### 2.1. Design Approach

We used a user participatory design approach for the design and development of FoVi [40,41]. The goal was to incorporate stroke survivors, their family members, and therapists into the design process to prioritize system capabilities, refine performance criteria, and clarify design requirements. We assembled an advisory team, including two occupational therapists (OTs), three physical therapists (PTs), and a user group comprised of five stroke survivors and their family members. The advisory team participated in focus group discussions throughout the design and development. After the final prototype was developed, the advisory team provided an initial evaluation of the device’s usability and its interfaces. To qualify for inclusion in the focus group, the OTs and PTs had to be 18 years or older and have at least three years of clinical experience working with stroke survivors. The stroke survivors had to be in the chronic phase of stroke recovery and medically stable to qualify for the focus group. The advisory team came to the research laboratory or participated remotely in up to six meetings per year based on the progress to discuss the design and development of the FoVi for up to 2 h per meeting. We collected formal consent from the therapist and stroke survivors before the first focus group. The study was approved by the University of Oklahoma Health Sciences Center Institution Review Board (IRB #9686).

### 2.2. Design Requirements

Our over-arching goal was to design a personal, wearable device that delivered targeted focal vibration to desired muscles safely and allowed a therapist to monitor and modify the treatment remotely. We divided our design goals into three major categories: wearability, vibration technology, and software interface. Software interface and vibration delivery requirements can be relatively precisely defined. In contrast, wearability requirements can be more fluid and open to interpretation. We have based our wearability requirements on relevant literature [42,43,44,45].

In terms of wearability design requirements, comfort, safety, ease of use, and customizability were the primary concerns. Based on the literature on wearability, we have defined them in Table 1 below.

Regarding vibration technology requirements, the main concern was effective vibration delivery. Therefore, we needed to meet or exceed the following criteria:(1)Individualized shape molding around the whole arm with six vibration motors to deliver vibratory stimuli in six different muscle locations.(2)Each vibration motor delivers a frequency between 60–300 Hz and amplitude between 0.1–10 mm.(3)Controllable activation and deactivation of each vibration motor and adjustable vibration parameters.(4)The FoVi should be comfortable, easy to wear, affordable, and flexible.(5)Easy to use interface for therapists to track the device usage and remotely adjust the vibration intensity and dosage. Based on rehabilitation progress, the therapist needs to change the settings to enable optimum recovery for the stroke survivors.(6)Rechargeable, with a battery lasting for at least 30 min.

The software design criteria were that the data collection should run as a background program, called a “service”, in the Android system. The user interface is a mobile application that accesses the vibration motors from this background service. The mobile app features include automatic runs to increase convenience for users. The App should display the vibration information, including which vibration motors are active and the frequency and amplitude of the vibration. The app should also monitor and record the vibration information in the background without affecting other smartphone functions such as calling, texting, etc. In addition, it should provide safety warnings and reminders to stroke survivors if the prescribed regime is not followed. The app should record the usage information, warnings, and reminders to a cloud service. A website portal should present the therapist information about FoVi usage, user compliance rates, and the number of safety warnings and reminders. As a wearable device to be used at home, the app interface for the user to interact with the device is critical. A web portal interface is convenient and important for a device to be remotely set up and monitored/adjusted by the therapist. We used the Insight^TM^ platform available through the Mobile Health (mHealth) Shared Resource at the Stephenson Cancer Center to customize, build, test, and launch the phone app and the web portal.

## 3. Results

### 3.1. Iterative Design Process

The five stroke survivors aged, on average, 62.2 years old (standard deviation: 12.4 years, range: 50 to 81 years old). They were all male; three had the stroke more than five years ago, and two had the stroke less than five years but more than two years ago. Three stroke survivors had right-side paresis, and two had left-side paresis. Changes were made throughout the development phase as more insights were gained from the advisory team. The first change occurred with the vibration motor pod. The original requirement was that each vibration motor pod should be independent in its electronics, rechargeable, and wirelessly controlled (Figure 1a). After the prototype was developed and tested, we decided to change the requirement so that each independent electronics box (wireless, rechargeable) would control four motors wired directly to it (Figure 1b) based on feedback from users and therapists. The feedback received was that the pods were too bulky when each pod had its electronics and battery included with the vibration motor. The general feedback about the bulkiness was that the parts’ thickness be minimized to the best extent as technologically feasible. The ideal goal was to match the thickness of the Myovolt vibration pods, which were 19 mm thick. Our original design with the electronics, battery, and vibration motor combined was 28 mm thick. In our revised design, the electronics box was 17 mm thick; the vibration motor pods were 10 mm thick. Having a standard electronics box (wireless controller, rechargeable) drive multiple motors reduced the bulkiness of the system. We revised the design to two electronic boxes each for four motors because this system still maintained more manageable and relatively short wire lengths.

The second design change was an addition of the option of a non-smartphone, microcontroller-based wireless remote with the capability of compliance monitoring. The second change was based on a discussion session where most stroke survivors mentioned wanting a remote, at least as an option. One survivor said the difficulty was in using a touch screen with cold fingers. It should be reliable, keep track of date/time, and be rechargeable, and the device’s firmware should be the same as the phone-based version. This requirement came after realizing that many stroke survivors may not be smartphone users or may not be comfortable using an app to interface the device.

The third design change involved the washing of the device “sleeve”. The third change was based on feedback from one focus group member who said he would be comfortable wearing the device repeatedly if it could be washed regularly, citing potential accidents where the fabric could get dirty, e.g., from a food spill. The original requirement was that the device is enclosed in a wearable sleeve that would be discarded or washed after the participant returned the device. The user could use disinfectant spray during use, but the device should be removed from the sleeve by only the engineers or therapists. The revised requirement is that the users should be able to remove the device from the sleeve. This additional requirement came after the last focus group meeting.

### 3.2. Vibration Pod Design

The vibration pods house 3V eccentric rotation mass (ERM) motors (14 mm length, 6 mm diameter). The motor has a rated 14,300 revolutions per minute (RPM) and typically normalized vibration amplitude of 4.51 g. At full power, the motors consume approximately 80 mA (130 mA peak) of current. The motor’s output frequencies range 80–170 Hz, with amplitudes in the range of 0.5–2.3 g with peak efficiency, which was ideal for our application. The pods are in a disk shape (28 mm diameter, 10 mm thickness). The pods are 3D-printed from polylactic acid (PLA) and sealed using epoxy. The size of the pods can be customized and was determined based on the size of the motors and preferences from the advisory panel.

### 3.3. Electronics Design

The core of the hardware is comprised of a custom printed circuit board (PCB) (55 × 30 mm) with a Microchip ATmega328 microcontroller (3.3 V, 8 MHz), as shown in Figure 2. The microcontroller can independently control up to four vibration motor pods for 500 mA through n-channel metal-oxide-semiconductor field-effect transistors (MOSFETS). The 500 mA battery will last 57 to 93 min, with all four motors running at full power.

Up to 15 different motor vibration intensities can be set independently for each vibration pod motor. The system is powered by a rechargeable, single-cell, 500 mA battery. The recharging circuit and indicator light-emitting diodes (LEDs) are designed into the PCB. Charging is performed through a micro-USB connector. The microcontroller monitors battery voltage, and a status LED is used to indicate when the battery is discharged. At this point, with the battery removed, the motors are turned off. During regular operation, the voltage of the Lipo battery changes as it gets discharged over time. This is considered when setting the pulse width modulation for the motors. The same vibration intensity is maintained regardless of the battery voltage.

Communication with the microcontroller is conducted wirelessly. There are two options for wireless communication. For an Android app over Bluetooth BLE 4.0, the PCB uses a Nordic nRF51822 Bluetooth module running opensource firmware by Adafruit Industries. The module relays UART serial data (9600 bps baud) over Bluetooth (Nordic UART GATT service) and vice versa. The PCB uses an XBee Series 3, 2.4 GHz, radio module for the wireless remote. The XBee Series 3 relays UART serial data (9600 bps baud) over the Digi XBee3 802.15.4 protocol and back. The PCB is designed such that either module can be soldered to it. This allows for a single PCB design for both the Android phone app and the remote-controlled versions of the device. The communication packets are the same for both versions, so the same firmware is used for both versions of the device. The hierarchy of communication is illustrated in Figure 3. Each slave device node has its address. A command packet for a slave device node contains commands for all four motors connected to it.

### 3.4. Ergonomics and Sleeve Design

The device is split into two sleeves with embedded motor pods and electronics boxes. Each sleeve has one control electronics box, which controls all the motor pods within that sleeve. All the components are put inside pockets, as shown in Figure 4. The forearm sleeve design is the same as the upper arm sleeve design. We made only one sleeve for each iteration for prototyping, which was used interchangeably for the upper arm and forearm. We are currently ordering final production versions of the sleeves out of Micromodal fabric. The users can remove the pods and the control box if they want to wash the sleeve. The sleeve is made of Micromodal material, which is breathable, moisture-wicking, and comfortable for close contact with the skin. Each sleeve is a loose fit for easy pulling on and pulling off a mostly non-functional upper limb by using a single hand. Embedded straps are looped around the motor pod. After pulling on the sleeve, these straps are closed and tightened, securing the sleeve after being pulled on. The straps are made of Velcro and are slightly elastic. This makes them easy to tighten. The straps have been designed for one-handed operation. The end of the strap can be strapped around whole, holding the start of the strap down with the thumb.

Figure 5 shows the forearm sleeve with the straps closed and sequence for one-hand operation of the sleeve. Some members of the focus groups suggested using Velcro hook–and–loop cinch straps, like hook–and–loop Velcro straps on some shoes. However, practically, that idea did not work out. Our in-lab testing showed that pulling on a hook-and-loop cinch strap caused it to rotate around the limb. Therefore, it offers no advantage over a simple strap in this application.

### 3.5. Interface and Webportal

As a wearable device to be used at home, the interface for the user and therapist to interact with the device is critical. For FoVi, based on feedback from stroke survivors, their family members, and therapists, we developed two interfaces for the users and a web portal for the therapist to manage, monitor, and track the usage of the device.

#### 3.5.1. User Interface (Smartphone App via Bluetooth)

The first user interface is a smartphone app to interact with the device. Simplicity is key, as indicated by the participants (as shown in Figure 6). Protocol and scheduling information are downloaded to the phone when the user is set up. When it is time for the vibration therapy to begin, the app generates a push notification on the smartphone (Figure 6a). The notification reminds the users that it is time for their vibration therapy session. Tapping on the notification launches the app, asking the user to tap the “go” button when ready (Figure 6b). At this point, the user puts on the device and taps the “go” button. The app asks for a confirmation. If the user taps “yes”, then the vibration therapy starts as the app sends the pre-assigned protocol commands to the device (Figure 6c). These run the designated focal vibration motor pods as assigned by the therapist.

The app logs all send commands and received acknowledgments with timestamps. These are uploaded to the Insight MHealth Platform whenever there is an internet connection. If there is no internet connection for the duration of the user participation, the uploading is completed at a follow-up visit after the user is done with the device.

#### 3.5.2. User Interface (Remote via Xbee)

Some of the participants and family members were not smartphone users or were not interested in using a smartphone to control the device. Therefore, a remote controller interface was developed. The remote comprises a Teensy 3.2 microcontroller (3.3 V, 96 MHz). Unlike an Android app where the phone is always on, this is only turned on when it is time for a session. Therefore, there is no reminder. The user has a printed schedule that they need to follow.

The remote has a start and stop button and an OLED (monochrome, 0.96-inch, 128 X 64 pixels, SPI) to display device status information. Once the start button is pressed, commands are sent to the sleeve electronics boxes according to the protocol, the same as the Android smartphone app. All transmitted commands and received acknowledgments are timestamped. The remote has an onboard Real Time Clock (Sparkfun DeadOn RTC Breakout DS3234), used to timestamp the command logs. Logs are recorded on a micro-SD card and on FRAM memory (MB85RS64V, 8KB) for backup to increase reliability. The logs are uploaded either from the SD card or over USB when the device is received back from the participant.

The remote is powered by a rechargeable LiPo battery (single cell, 3.7 V, 1200 mAh). A micro-USB charger (Sparkfun LiPo Charger/Booster—5 V/1A) is built into the remote. The remote is not connected to the online Therapist Interface (Section 3.5.3). Protocol commands are programmed directly into the microcontroller by the FoVi command generator, as shown in Figure 7. This graphical user interface (GUI) accepts desired motor commands from drop-down lists. It was developed in the Python programming language. The user can select the slave device node address and the settings for the various motors. An image of the slave node and motor pod locations helps identify the correct slave node and motor pod. Since the protocol for both Bluetooth and Xbee radio is the same, the same encoded commands can be used for both app and remote interfaces.

#### 3.5.3. Therapist Interface (Web Portal)

The therapist interface is a web-based portal used to set up focal vibration therapy protocols, create user profiles, and assign therapy protocols and scheduling to users (Figure 8). This is based on the Insight mHealth Platform [46]. A therapy protocol consists of vibration sessions with specific motors, with optional breaks in between. For example, a therapy protocol with three vibration sessions can be as follows:Set slave 0 motors 0 and 1 to 80% intensity, and set all others to off (for 5 min);Set all motors to off (for a 5 min break);Set slave one motors 0, 1, and 3 to 50% intensity, and set all others to off (for 5 min);Set all motors to off (for a 5 min break);Set slave 0 motors 0 and 1 to 80% intensity, and set all others to off (for 5 min);Set all motors to off (end of session).

User scheduling is defined as the assignment of the protocol to a user over a calendar period. For example, a user can schedule a protocol for 10:00 a.m. every Monday, Wednesday, and Friday, for four weeks. Reminders can also be set, e.g., 30 min before the scheduled time. The therapist interface also shows device usage logs automatically uploaded by the app over a secure protocol or when uploading the usage from the micro-SD card in the remote controller. A log includes all the commands sent to the sleeves and the acknowledgment packets. All of these are time-stamped.

### 3.6. Initial Test and User Evaluation

We measured the vibration characteristics at the low (6.9 g), medium (13.0 g), and high (19.1 g) settings. The results are presented in Figure 9. We used an accelerometer (STMicroelectronics LSM9DS1 ± 16 g’s full scale, 952 Hz sampling rate). For details on how we computed amplitude in mm using accelerometer data, please see the authors’ previous work on vibration parameters of effective wearable devices [47].

We tested the device battery (single-cell, 500 mAh LiPo) life. The battery was fully charged, and three focal vibration pods were run at the “high” intensity setting until it dropped down to the lower voltage level (3.3 V). The battery lasted 3 h and 40 min. This was more than adequate for our 30 min battery life requirement.

We tested the device with therapists and obtained their feedback. A summary of their feedback is presented in Table 2. As for the usability and feasibility of five stroke survivors, they completed a 10-min vibration protocol without any adverse event or negative feedback. Open-ended questions regarding the experiences of all participants were summarized in Table 3. Regarding their comment on providing more evidence-based settings, this will be our future work, as currently, there is not enough evidence to provide such a feature.

## 4. Discussion

In this paper, the authors presented a user participatory design approach to design, develop, and evaluate a wearable focal vibration device, FoVi. The main functions of FoVi are to deliver targeted focal vibration to desired muscles safely, to log usage information of the vibration, and to enable real-time connection between stroke survivors and therapists. The vibration will be programmed in intensity, duration, and interval of actuation or can be continuous. With the prototype developed, all the main functions were achieved, and therapists were able to adjust each motor’s intensity, duration, and interval for highly customizable vibration. All the participants perceived well the user participatory design approach, and the authors recommend that the approach be adapted for all rehabilitative and assistive technology development. We measured user perception indirectly. The participants were enthusiastic about participating and gave a lot of thought to their points in discussions. Several participants commented that they were glad we discussed the design with them before rather than after. The user participatory design approach has several advantages. It ensures the needs and requirements of the stroke survivors, and therapists are addressed and incorporated into the design. It provides a rich set of design feedback early in the concept, potentially saving a lot of expense and time spent on design revisions. It can highlight unanticipated use-case scenarios and hence make the design even better. Users and therapists have more confidence in using the device if they know that other users and therapists were involved in its design. On the other hand, the user participatory design approach can be time-consuming since meetings must be scheduled with multiple stakeholders, and the participatory sessions can take a lot of time. The approach can be challenging to predict since the outcome and content of participatory sessions are unknown.

Device tolerability is an important determinant of subject adherence and intervention effectiveness. Wearable textiles have been described as feeling “uncomfortable”, “annoying”, and “numbing” [48]. Other challenges reported include limited size and thickness, the use of electronics in moist environments, and visibility of the textile. With the user participatory design approach, we could minimize some of the challenges in the iterative design process. During the final usability assessment, both therapist and stroke survivor participants reported a more positive experience with almost no negative experience with our prototype. These findings agreed with another study on stroke survivors who reported significantly more positive than negative effects with focal vibration [24]. The training and instructions allowed participants to experience what a vibration treatment session would be like and get familiar with the interfaces. In addition, the participant would only need to wear the system for the vibration treatment, which the therapist and users commented was an important feature for the home-based application of FV. The authors offered the participants trials with MyoVolt during the focus groups and final evaluation. Compared to the MyoVolt, the participants liked the FoVi sleeve, which was more comfortable and easier to wear and take off with one hand. They commented that the MyoVolt was more professional, appealing, and had higher-quality manufacturing. In future studies, we will improve the appeal of FoVi and test it for a longer time, such as four weeks at home.

One noticeable limitation of the current FoVi prototype and other available wearable FV devices is the lack of feedback to provide accurate vibration, especially when the vibration motor attaches to the human body with varying tightness. The vibration intensity (associated with vibration frequency and amplitude) was determined by the controller output for free vibration. When the vibration motor is attached to the human body, depending on the tightness, the actual vibration amplitude will be different than the free vibration. We did not monitor the amplitude when the device was worn in the current prototype, and there is no mechanism to adjust the amplitude. In our subsequent development, we plan to integrate an accelerometer to measure the vibration amplitude and add a feedback control to adjust the amplitude in real-time based on the tightness of the sleeve.

## 5. Conclusions and Future Research

The primary contribution of this paper was to introduce the user participatory design approach to researchers and therapists when designing and developing rehabilitative and assistive technologies. The involvement of stroke survivors, family members, and therapists during the whole design procedure will increase the success of the development and ensure the acceptability of the final product. The developed FoVi final prototype met all the design requirements and was well perceived by stroke survivors, their family members, and therapists on the device and its interfaces. The therapists liked the app and web portal interfaces to set up, monitor, and adjust the vibration prescriptions based on rehabilitation progress. The stroke survivors liked the sleeve for its ease to wear and take off and the choices of a smartphone app interface or a remote controller to operate the device. Lastly, the developed FoVi as a smart connect technology could provide stroke survivors a sustainable home-based intervention and allow therapists to track the compliance and the progress of the rehabilitation. Future research and development include data collected by these devices that can enable therapists to create datasets that, in the future, using machine learning algorithms, can allow researchers to work on the identification of optimal rehabilitation patterns.

## 6. Patents

A patent had been filed and is currently pending: Wearable Focal Vibration Device and Methods of Use (2020), Patent pending: 17/206,016.

## Figures and Tables

**Figure 1 sensors-22-03308-f001:**
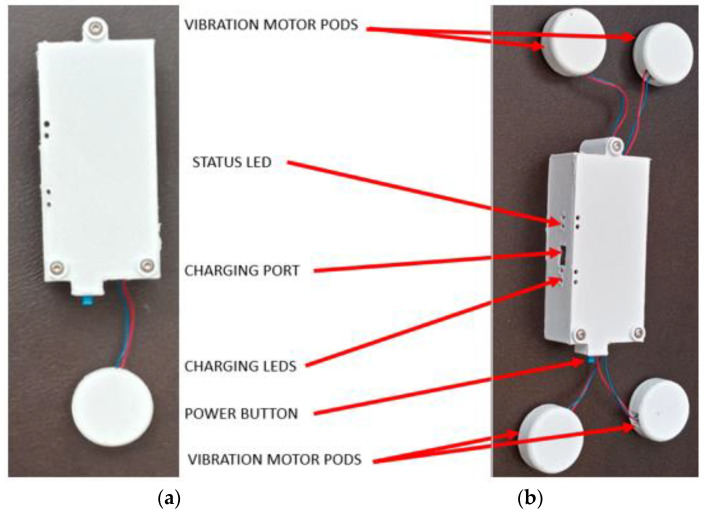
Prototype of FoVi. ((**a**)—left) The first prototype with one pod attached to the main controller and ((**b**)—right) the final design with four pods attaches to the main controller.

**Figure 2 sensors-22-03308-f002:**
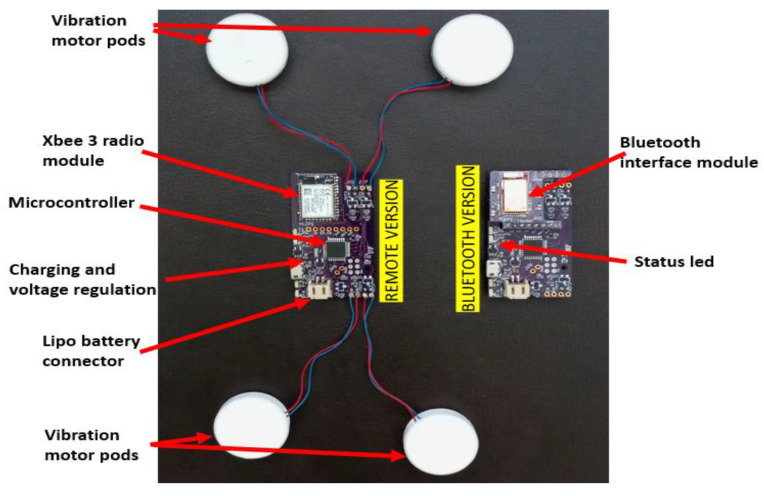
Electronics design of FoVi. Elements of the control electronics box are shown with connected motors. All the elements are on a custom-printed circuit board.

**Figure 3 sensors-22-03308-f003:**
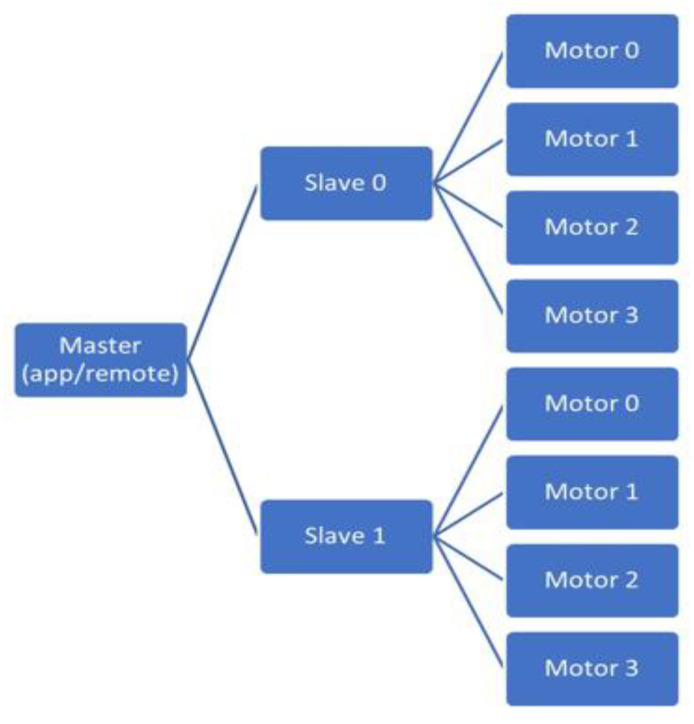
FoVi communication hierarchy. The “master” (app or remote) communicates to two “slave” devices: one in the upper arm sleeve and the other in the forearm sleeve. Each of these “slaves” in turn control up to four focal vibration motor pods.

**Figure 4 sensors-22-03308-f004:**
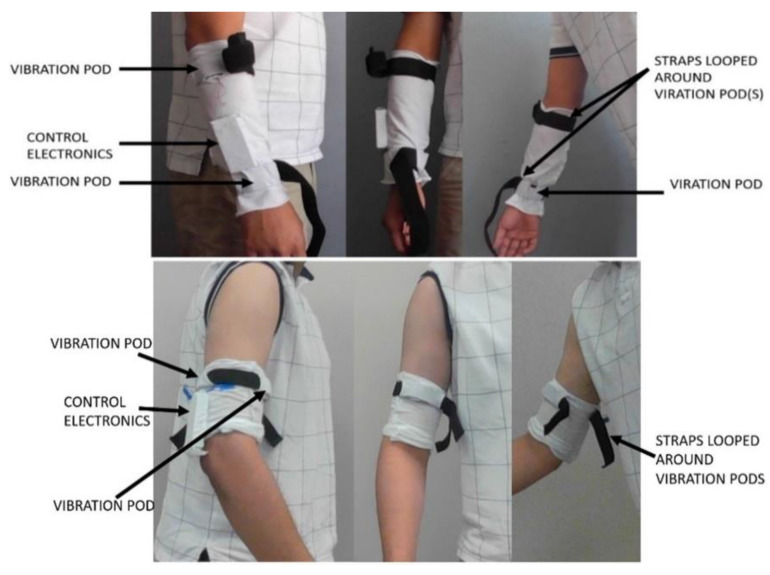
Top: Forearm sleeve with straps open. Embedded focal vibration pods and a control electronics box are labeled. These can be removed if the sleeve needs to be washed. Bottom: when the vibration pods and control electronics are attached to the upper arm with the sleeve.

**Figure 5 sensors-22-03308-f005:**
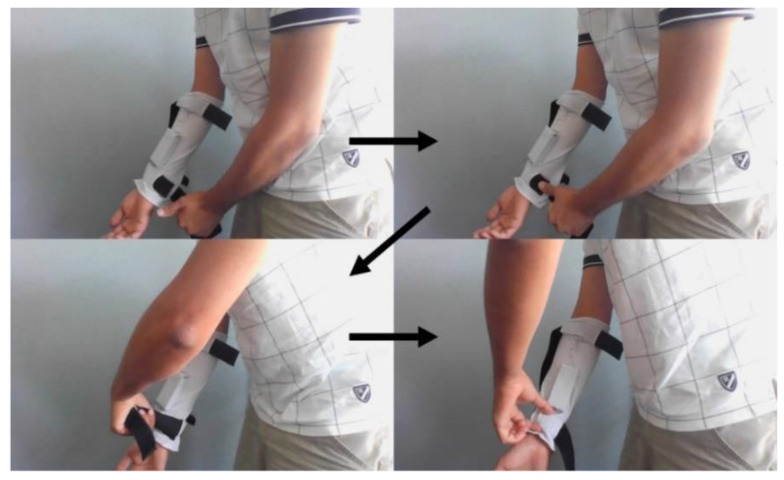
The sequence of one-handed closing of the straps. First, the end of the strap is gripped in the hand. Next, the start of the strap is held down with the thumb. While holding this down with the thumb, the strap end can be looped over and closed. Finally, the vibration pod location within the sleeve can be adjusted.

**Figure 6 sensors-22-03308-f006:**
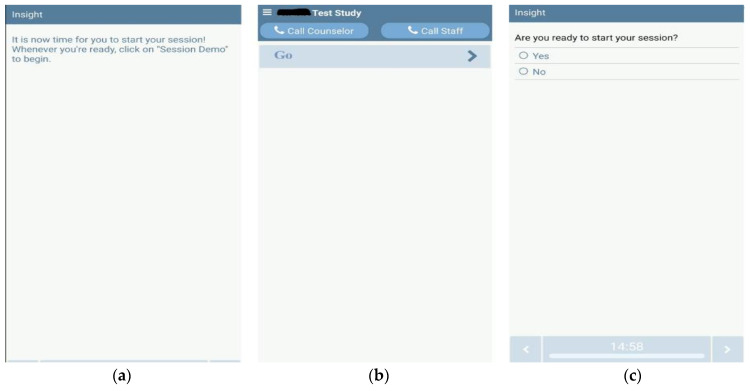
User interface over the Android Bluetooth app. Left—(**a**) The user receives a push notification reminder, which takes them to the app. Middle—(**b**) The “Go” button for the users to start the vibration therapy after they receive the notification. Right—(**c**) If confirmed by yes, the prescribed vibration therapy protocol starts running the focal vibration pods. This helps to prevent accidental activation.

**Figure 7 sensors-22-03308-f007:**
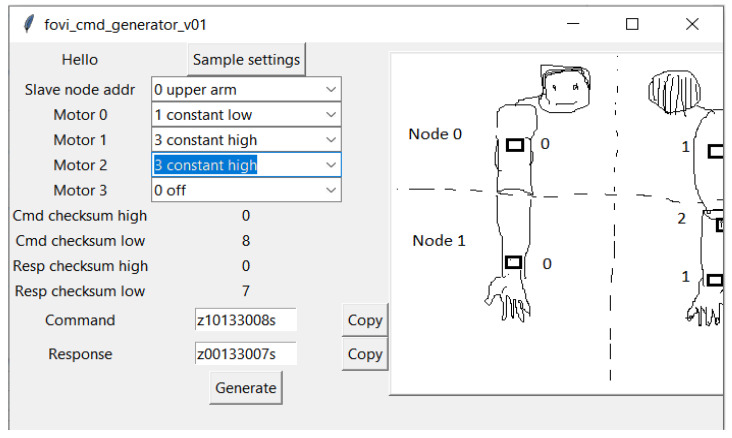
FoVi Command Generator. Graphical user interface for generating encoded command packets. The therapist can select the slave device node and the setting for each of the four vibration motor pods. An illustration of their locations is included for easy identification.

**Figure 8 sensors-22-03308-f008:**
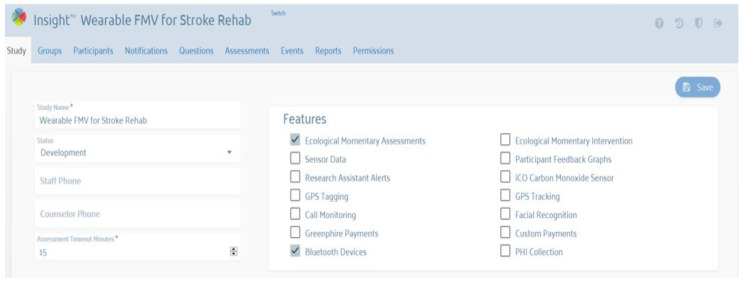
Web portal for the therapist to set up the device for home-based vibration therapy. The therapist can integrate the vibration therapy with feedback surveys after FV therapy and even real-time surveys, e.g., during FV therapy.

**Figure 9 sensors-22-03308-f009:**
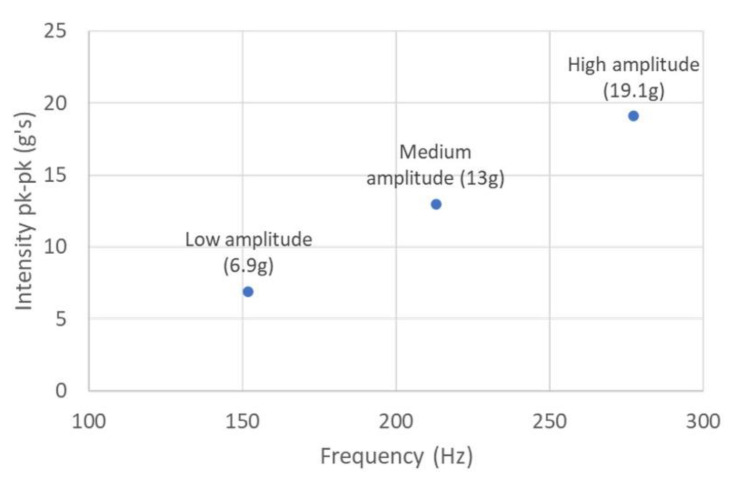
For the low setting, 6.9 g’s peak-to-peak computes to 0.07 mm peak-to-peak amplitude. For the “medium” setting, 13 g’s peak-to-peak computes to 0.07 mm peak-to-peak. For the “high” setting, 19.1 g’s peak-to-peak computes to 0.06 mm peak-to-peak.

**Table 1 sensors-22-03308-t001:** Design requirements for the wearability of FoVi.

**Requirements**	**Definition**
Comfort (shape)	Acceptable temperature, texture, shape, and tightness [43,45].
Comfort (ease of use)	Easy to set up and control for users with different needs [42,45].
Comfort (usability)	Easy to put on and take off, considering full use of one arm and very little mobility on the affected arm [44,45].
Comfort (obtrusiveness)	Enable and not hinder the natural body movements and functions [45].
Comfort (sizing)	Customizable to fit different users’ sizes or sides (right or left) [42,43,44,45].
Safety (harm)	Should not cause harm or pain [43,45].
Safety (security)	Should attach securely with no risk of the device coming off. [44,45].

**Table 2 sensors-22-03308-t002:** Therapists’ feedback on the FoVi prototype, app, and web portal (SD = Strongly disagree, D = disagree, N = neither agree nor disagree, A = agree, and SA = Strongly agree).

Questions Asked to the Therapists	Responses (# of the Therapist)				
	SD	D	N	A	SA
Q1. Would like to use the app and web portal to monitor the usage of FoVi	0	0	0	1	4
Q2. The app and web portal are easy to use	0	2	0	3	0
Q3. May need a technical person while using the device long-term	0	0	0	5	0
Q4. Most people will use the device quickly	0	1	0	0	4

**Table 3 sensors-22-03308-t003:** Participants’ comments and suggestions for FoVi.

Source (# of Participants)	Comment/Suggestion
Therapist (5)	Concerned about the storage of data and reliability of web portal.
Therapist (5)	Integrate the web portal into the system used by the hospital.
Therapist (5)	Use some common selection of frequency and amplitude settings instead of individually adjusting each parameter.
Therapist (5)	Provide more evidence-based settings for users with different levels of severity and functional capacity.
Therapist (1)	User app could be challenge for less smartphone experience.
Stroke survivors (5)	Liked the feel of the vibration.
Stroke survivors (5)	The feature for adjusting vibration was very helpful.

## Data Availability

The dataset can be shared with the approval from OUHSC IRB and the PI of the study (H.W. who is the corresponding author for this article). Requests to access the data should be directed to H.W., Hongwu.wang@phhp.ufl.edu.

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
