# Peer review of "User Participatory Design of a Wearable Focal Vibration Device for Home-Based Stroke Rehabilitation"

_sensors, 2022, doi:10.3390/s22093308_

Round 1

Reviewer 1 Report

This interesting paper presents the user participatory design process of a wearable focal vibration device for home-based stroke rehabilitation supervised remotely by professional therapists.

I firmly believe that this design approach can be very suitable for a product like this because therapy adherence and efficacy is highly influence by both patients and therapists user experience.

Main strengths of this research work are the design approach and the results section. However, I recommend improving the Introduction, Design Requirements, Results and Discussion sections.

Below I list and explain my comments. I hope they help you to improve your paper:

  • Update and improve references (Lines 42 to 54): In line 42 the authors explain that numerous therapies have been developed over the last 10 years to treat acute ischemic stroke but the references used to justify it are from 2011 and we are in 2022. Please, include more recent references that can be really representative for these last 10 years.
  • The authors use the word “sustainability” in line 51 and 73 but this concept seems to be very ambiguous for the reader. Reference [11] related to barriers to implementation of rehabilitation treatments refers several factors such as lack of time, staffing issues, training/education, therapy selection and prioritization, equipment availability and team functioning / communication but all these issues refer to medical teams. The authors should explain the term sustainability in the context of their research and what are the factors are included in this term in relation to both patients and therapists.
  • In section section 2.2 talking about design requirement of the wearable the authors refer to the concept of “comfort” several times (line 120, line 129 and line 135. A previous review of the state of the art with some relevant references about design requirements for wearables is recommended, because concepts such as this of “comfort” are more complex and should be carefully defined. Please, review references such as the following one: Francés-Morcillo, L., Morer-Camo, P., Rodríguez-Ferradas, M. I., & Cazón-Martín, A. (2020). Wearable design requirements identification and evaluation. Sensors, 20(9), 2599.
  • In section 3.1 the authors describe the iterative design process. Some of the design changes are exposed but the insights achieved in the participatory design process, on which these design changes are based, are not always explained. For example, why the authors took the decision to group four motors with one electronic box instead of that each motor pod should be independent electronically? (Lines 162 to 167).
  • In the description of Figure 8 of section 3.5.3 related to Therapist Interface, the authors explain that the therapist can integrate the vibration therapy with other assessments such as the ecological momentary assessment. This sentence is difficult to understand, please explain it in a better way.
  • In section 3.6 from line 357 to 375 the results of the initial test and user evaluation are presented. The way of presenting these results is very confusing. I suggest to present these results using some graphics (for quantitative results) and tables (for qualitative results) to make it easier to understand to readers. Please, include the questionnaires used to collect feedback from both users and therapists as an annex to this article.
  • In section 4 from line 384 to 386 the authors explain that the user participatory design approach was well perceived by all participants. How have the authors measured this perception? This section should be improved to discuss the strengths and weaknesses of the proposed design method.
  • In section 4 line 389 the reference [36] is correct? This reference corresponds to the website of a wearable but this website does not include information on the aspects mentioned in the paper related to device tolerability.
  • It may be a good idea to rename section 5 as "Conclusions and future research" and include an idea for future research development based in the data collected by these devices that can allow therapists to create datasets that in the future, using machine learning algorithms, can allow researchers to work on the identification of optimal rehabilitation patterns.

I hope these comments will help you to improve your paper. All the best in the review of this paper.

Author Response

Thank you very much for your comments and suggestions; please see the attached file with our detailed responses and the revised manuscript.

Reviewer 2 Report

This manuscript presents both focus group input and device designs for a focal vibration device.

I really like the concept for this paper, combining the focus group feedback with device design and iterative processes. It highlights some aspects that are missing from a lot of device design papers. 

There are two key elements that I think could be substantially improved. 

1)  Include more results from the focus groups themselves and not just in the context of how it changed design elements. Summaries of their input in tables or in some other format would be valuable to the reader. Elements are presently included in relation to design changes, but it would be useful to know more about the nature of the questions that were asked and themes in people's responses. It would also be really helpful to understand how the focus group influenced the input requirements. 

2) Is the intention of this paper to enable other researchers to build their own devices? If so, including supplementary materials with schematics, code, and models would be useful and significantly increase the value of this manuscript.

Clarification/presentation comments:

1) Stroke survivors are not patients in this context. I recommend not referring to them as patients here. 

2) Line 48 could be misinterpreted to about inability to activate muscles at all rather than ability to activate muscles independently. 

3) While under-dosage may be a contributing factor, is it really the primary limitation? Will everyone improve? 

4) Are the lack of uniform protocols and sustainability barriers to clinicians prescribing the device or stroke survivors using the device? What do you mean by sustainability in this context?

5) What are you referring to in terms of "participant memory" in Lines 88-89? Is it that the user needs to remember how to use the device? 

6) Why is device documentation important? I'm assuming it's so that recovery and compliance can be tracked, but it would be useful make this explicit. 

7) What do you mean by "consistent vibration" in Line 92? Is it that the device varies in functionality or that the motor/skin interface changes the effect on the sensory organs?

8) Line 130 - why does the therapist need to be able to track usage and adjust the settings?

9) The manuscript explains that there was a change from 6 motors to 4, but it isn't clear why that change happened. 

10) Why is the therapist's interaction with the device critical? 

11) The device is described as being intended for the arm, but it looks like it is only applied to the forearm. This should be clarified. 

Author Response

Thank you very much for your comments and excellent suggestions. Please see attached file with our detailed responses and the revised manuscript.

Round 2

Reviewer 2 Report

The manuscript has improved with the revisions. 

If references 42-45 are the same ones that are applied in Table 1, I'd recommend including relevant references with the lines in the Table. 

When the participants gave feedback about the size of the prior iteration (e.g. too bulky), was there any back and forth that provided insight about a reasonable size? This type of content would be useful to include. 

Please include an image of the sleeve for the upper arm as well as the forearm. 

Author Response

If references 42-45 are the same ones that are applied in Table 1, I'd recommend including relevant references with the lines in the Table. 

Thank you for the suggestion and we added the relevant references with the lines in Table 1.

When the participants gave feedback about the size of the prior iteration (e.g. too bulky), was there any back and forth that provided insight about a reasonable size? This type of content would be useful to include. 

Thank you for the suggestion. We added the below content to the revision:

The general feedback about the bulkiness was that the parts’ thickness be minimized to the best extent as technologically feasible. The ideal goal was to match the thickness of the Myovolt vibration pods, which were 19 mm thick. Our original design with electronics, battery, and vibration motor combined was 28 mm thick. In our revised design, the electronics box was 17 mm thick; the vibration motor pods were 10 mm thick. 

Please include an image of the sleeve for the upper arm as well as the forearm. 

Thank you for the suggestion. We added the upper arm sleeve image to Figure 4.
